# Filtering Matters: Experiments in Filtering Training Sets for Machine Translation

**Steinþór Steingrímsson[1], Hrafn Loftsson[1], and Andy Way[2]**

[1]Department of Computer Science, Reykjavik University, Iceland
[2]ADAPT Centre, School of Computing, Dublin City University, Ireland
steinthor18@ru.is, hrafn@ru.is, andy.way@adaptcentre.ie

## Abstract

We explore different approaches for filtering parallel data for MT training, whether the same filtering approaches suit different datasets, and if separate filters should be applied to a dataset depending on the translation direction. We evaluate the results of different approaches, both manually and on a downstream NMT task. We find that, first, it is beneficial to inspect how well different filtering approaches suit different datasets and, second, that while MT systems trained on data prepared using different filters do not differ substantially in quality, there is indeed a statistically significant difference. Finally, we find that the same training sets do not seem to suit different translation directions.

## 1 Introduction

In recent years, machine learning (ML) research has generally focused on creating better models rather than better datasets. The focus on benchmarking model performance spurs researchers into adapting the largest existing datasets without fully considering fidelity to the underlying problem the model should solve (Mazumder et al., 2022). The effectiveness of ML models, however, depends on both algorithms and data. Aroyo et al. (2022) argue that as the datasets define the world within which models exist and operate, more work is needed on how the data can be optimized for more effective use.

Filtering parallel data for machine translation (MT) is the task of removing possible detrimental segments from data used for training MT models. Detrimental segments, in this context, are sentence pairs in the training data that may degrade the performance of an MT system trained on the data. Filtering is usually carried out using a set of rules, scoring mechanisms and/or classifiers, to remove sentence pairs with the lowest perceived quality.

In this work, we experiment with filtering the raw data from two parallel corpora, ParaCrawl (Bañón et al., 2020) and ParIce (Barkarson and Steingrímsson, 2019), for the English–Icelandic language pair. Our goal is to minimize detrimental data while losing little or no useful data from the texts, thus building a more accurate training set.

We investigate how shallow filters, four different scoring mechanisms and different classifiers based on them are suited to score and filter English–Icelandic sentence pairs. We compare these to Bicleaner (Sánchez-Cartagena et al., 2018; Ramírez-Sánchez et al., 2020) and to how the two corpora (ParaCrawl and ParIce) were filtered for publishing.

Recent literature on parallel corpus filtering has largely focused on filtering noisy data collected from the web, as discussed in Section 2. We want to investigate whether the same approaches are suitable to filter noisy web-crawled corpora and cleaner parallel corpora compiled from document pairs that are known to be mutual translations. Furthermore, although the same training data is usually used for training both translation directions, source→target and target→source, for a given dataset, we investigate whether that is optimal or whether filtering separately for each translation direction is likely to bring improvements to a downstream MT task.

Our primary goal is to find out how to filter parallel corpora so as to compile a training set that potentially gives the best results when used to train an MT system. We seek an answer to the following research questions: **1) Should the same filtering approaches be used for a given language pair, regardless of the datasets being filtered? 2) Should the intended translation di-**

**rection of an MT system effect how the data, used to train the system, is filtered?** In order to answer these questions, we build MT models for both translation directions and multiple different filtering approaches for each one, and evaluate the results, both manually and automatically. We find for best results, specific filtering approaches should be chosen based on the dataset and translation direction being filtered.

## 2 Related Work

In their paper, Khayrallah and Koehn (2018) show that incorrect translations, untranslated target text, misalignments, and other noisy segments in a parallel corpus have a detrimental effect on the output quality of NMT systems trained on that corpus, as measured using BLEU (Papineni et al., 2002). They specify five general classes of noise commonly found in the German-English ParaCrawl corpus: misaligned sentences, disfluent text, wrong language, short segments, and untranslated sentences. As this classification is rather coarse, some variation can be expected within each class; a misalignment in one sentence pair does not have to be equivalent to a misalignment in another sentence pair.

Briakou and Carpuat (2021) focus on fine-grained semantic divergences within mostly equivalent pairs (pairs of words, phrases or sentences that have similar meanings and connotations). An example given in the paper is fr: "votre père est français" → en: "your parent is french", where the correct translation should be: "your father is french". These fine-grained divergences can even be found in high-quality parallel corpora. They find that the divergences cause degradation on the MT output of a system trained on the data, as measured by BLEU and METEOR (Banerjee and Lavie, 2005), and that divergences impact model confidence in their predictions. Lexical substitution causes the largest degradation and subtree deletion the least. Nevertheless, the impact on divergences seem to be smaller than that of noise. They argue that this suggests that noise-filtering techniques are subobtimal to deal with fine-grained divergences.

In early work on filtering web-scraped parallel corpora, Rarrick et al. (2011) filter out machine-translated content and show that removing large amounts of training data can improve performance of an MT system, challenging conventional wisdom at the time that more data is better.

Cross-lingual word embeddings have been used to calculate distance between equivalences in different languages (Luong et al., 2015; Artetxe et al., 2016). Defauw et al. (2019) treat filtering as a supervised regression problem and show that Levenshtein distance (Levenshtein, 1966) between the target and MT-translated source, as well as cosine distance between sentence embeddings of the source and target, are important features.

The Conference on Machine Translation, WMT, hosted three annual shared tasks on parallel corpus filtering (Koehn et al., 2018, 2019, 2020), focusing on filtering noisy web-crawled corpora. Chaudhary et al. (2019) and Artetxe and Schwenk (2019a) introduced approaches based on cross-lingual sentence embeddings trained from parallel sentences. When using cosine similarity to find the nearest neighbours in an embedding space, cosine similarity is not necessarily globally consistent and different scales of target candidates for a given source sentence may affect their relative ranking, causing the hubness problem, described by Dinu and Baroni (2015). The problem is caused by a few vectors in the embedding space that tend to be "universal" neighbours, i.e., neighbours of a large number of different mapped vectors, pushing the correct ones down the neighbour list. Both papers tackle the scale inconsistencies of cosine similarity by considering the margin between a given sentence pair and its closest candidates to normalize the similarity scores.

Bicleaner uses a set of handcrafted rules to detect flawed sentences and then proceeds to use a random forest classifier based on lexical translations and several shallow features such as respective length, matching numbers and punctuation. It also scores sentences based on fluency using 5-gram language models. Bicleaner AI (Zaragoza-Bernabeu et al., 2022) is a fork of Bicleaner using a neural classifier. It has been shown to give significant improvements in translation quality as measured by BLEU when used for filtering training data for multiple language pairs, as compared to the previous version of the tool. In contrast to tools that implement a single method for parallel corpus filtering, Aulamo et al. (2020) implement multiple different filters in the OpusFilter toolbox. These include heuristic based filters, language identification, character-based language models and word alignment tools.

Herold et al. (2022) revisit the noise classes specified by Khayrallah and Koehn (2018) to investigate how accurately two of the strongest filtering approaches to date (according to them) cross entropy (Rossenbach et al., 2018) and LASER (Artetxe and Schwenk, 2019b) can filter out different noise classes. They find that for a common language pair, German→English, most types of noise can be detected with over 90% accuracy, although misalignments and poor synthetic translation can only be detected with an accuracy of less than 70%. For a less common language pair, Khmer–English, the performance of the filtering system degraded significantly and the accuracy of identifying noise was lowered by 8–19%, depending on the type of noise.

## 3 Experimental Setup

We compare a number of approaches and scoring mechanisms and apply them to a web-crawled corpus, on the one hand, and a parallel corpus compiled from parallel documents, on the other. We manually evaluate samples of the results using the taxonomy developed by Kreutzer et al. (2022) to gain an understanding of what sort of data each approach and scoring mechanism filters out. We then train MT systems on datasets filtered using the different approaches, as well as on previously published, filtered versions of the corpora, and compare the quality of the resulting systems in terms of BLEU scores. We measure BLEU scores on the test set provided for the English–Icelandic language pair in the WMT 2021 shared task (Akhbardeh et al., 2021), using SacreBLEU (Post, 2018).

### 3.1 Data Sets

The data sets we use for our experiments are the English–Icelandic part of ParaCrawl and the English–Icelandic parallel corpus ParIce. We carry out the same experiments using both corpora and compare the results.

ParaCrawl is compiled from web-crawled data. Based on the evaluation by Kreutzer et al. (2022), approximately 76% of sentence pairs are acceptable mutual translations, on average, in 21 language pairs from the ParaCrawl 7.1 datasets cleaned for publication. There is also high variance between languages and low-resource datasets tend to have lowest human-judged quality. Rikters (2018) inspects the quality of the first version of ParaCrawl and filters out 85% of the English–Estonian ParaCrawl dataset. Although there may be differences in noise ratio between different versions of the corpus, for most language pairs ParaCrawl can likely be made more useful for training MT models by better filtering. This has been emphasized by the results of the WMT shared tasks on filtering parallel corpora. In our work, we start with the raw data from version 9 of the corpus, consisting of 65,373,727 sentence pairs in total. Our goal is to extract from the corpus sentence pairs useful for training MT systems on its own or to complement other data sets, and leave out sentence pairs likely to be detrimental.

The English-Icelandic parallel corpus ParIce differs from ParaCrawl in that it is compiled from known parallel documents, which have been aligned at the sentence level. When the corpus was compiled initially, the filtering process resulted in an estimated 20% reduction in corpus size. Out of what remained, manual evaluation of samples from the corpus indicated that approximately 3.5% was in some way faulty (Barkarson and Steingrímsson, 2019). The corpus is available unfiltered, accompanied with semantic similarity scores for each sentence pair and flags indicating whether it is recommended to filter out the pair or not. We work with the unfiltered data, version 21.10 (Steingrímsson and Barkarson, 2021).

### 3.2 Filters and Scoring

In order to find which sentence pairs are useful and which ones to filter out, we use an array of tools for scoring sentence pairs to find the highest-quality data within the corpora. We start with shallow filters to remove pairs that are very likely to be noise, and then proceed to run different tools, both made available by others and of our own device.

**Shallow Filters**
Our shallow filters are inspired by Pinnis (2018), who applies 17 different filters in his work. We do not use all his filtering approaches but select the ones likely to remove the highest portion of detrimental pairs as outlined by Khayrallah and Koehn (2018). They are:

1. If both source and target sentence have 3 tokens or less, the pair is discarded.

2. All pairs, for which 60% or more of the tokens in one language are also present in the other language, are removed.

3. At minimum, 70% of characters in both sentences should be alphabetical.

4. Both languages are in the top 2 prediction of a language filter. We use fasttext (Joulin et al., 2017) for language filtering.

5. Removal of near-duplicate pairs. We consider sentence pairs with all non-alphabetical symbols removed and if there are identical pairs in the corpus we keep the one with the highest score (Bicleaner score for ParaCrawl, LASER, LaBSE and WAScore for ParIce).

6. Removal of near-duplicate source or target sentence. We consider strings after removing all non-alphabetical symbols, and all tokens starting with a capital letter (removing possible named entities) from the sentences. If there are identical such strings in the same language, we select the highest scoring pair.

## Bicleaner Models

Bicleaner is an open source noise filter and classification tool to clean parallel corpora, released as part of the ParaCrawl project and used to generate the filtered ParaCrawl datasets. Bicleaner uses a set of hard rules for pre-filtering, n-gram models for fluency scoring, and a random forest classifier to produce a probability score using features such as lexical similarity, sentence length, punctuation and capitalization. Bicleaner AI is a fork that uses a fine-tuned XLM-RoBERTa classifier to produce probability scores by training it on positive samples from existing parallel corpora and negative samples which are created by corrupting the same positive samples. In synthesising the noise, the tool tries to emulate errors commonly introduced by sentence segmentation and alignment.

We use two publicly available Bicleaner models, version 1.5, for English–Icelandic, and Bicleaner AI 1.0 full model. In addition, we train two new models using Bicleaner v0.15.2, one that classifies lemmatized data and the other unlemmatized. For training each model, we used word frequency information from the Icelandic Gigaword Corpus (Steingrímsson et al., 2018) for Icelandic and News Crawl (Barrault et al., 2020) for English, a probabilistic dictionary (Steingrímsson et al., 2022), and for parallel training data, 250k highest-scoring sentence pairs from the 21.10 version of ParIce (Steingrímsson and Barkarson, 2021), based on the scores published with the corpus.

## Scoring and Score-Based Classifiers

We use multiple scoring mechanisms to assess the quality of the bilingual sentence pairs.

**LASER** (Artetxe and Schwenk, 2019b) uses a single BiLSTM encoder with a shared byte-pair encoding (BPE) vocabulary (Sennrich et al., 2016) for all languages and is trained on parallel corpora.

**LaBSE** (Feng et al., 2022) is trained and optimized to produce similar representations for bilingual sentence pairs. It uses dual encoder models, with the encoder architecture following the BERT Base model, and additive margin softmax which extends the scoring function in the model by introducing a large margin around positive pairs, improving the separation between translations and nearby non-translations (Yang et al., 2019). An available pre-trained model was trained on 109 languages, including Icelandic and English.

**NMTScore** (Vamvas and Sennrich, 2022) is based on translation cross-likelihood, the likelihood that a translation of segment *A* into some language, could also be a translation of segment *B* into the same language. An example could be the translation of the French 'Bonjour!' into the Swedish 'Hej!'. To calculate translation cross-likelihood, the French segment would first be translated to a third language, say English, and the score is based on the probability of the model getting the same translation for the Swedish segment. The score is symmetrized by averaging the translation probabilities in both directions. We use the M2M100 multilingual translation model (Fan et al., 2021) to calculate NMTScore.

**WAScore** is a word alignment-based score devised to measure word-level parallelism, introduced in Steingrímsson et al. (2021) to help with identifying parallel bilingual sentence pairs in a comparable corpus.

The scores are used to train classifiers for determining acceptability of parallel sentence pairs. We adapt a training set compiled for a classifier used in mining comparable corpora (Steingrímsson et al., 2021). The dataset was compiled of 50,000 randomly sampled non-parallel pairs from two monolingual news corpora for negative examples and 1,000 parallel segments containing sentence pairs from news media. LASER, LaBSE, NMTScore and WAScore were calculated for all 51K sentence pairs, and used to train the classifiers. We used scikit-learn (Pedregosa et al., 2011) to train random forest (Breiman, 2001), support

| ParaCrawl shallow filtering | | | | | |
|---|---|---|---|---|---|
| Filter | Dataset Size | CC (%) | 3C (%) | X (%) | 3X (%) |
| 0. ParaCrawl v9 Raw | 65,373,727 | 14.40 | 69.20 | 8.00 | 30.80 |
| 0b. ParaCrawl v9 Clean | 2,967,519 | 13.60 | 89.20 | 8.80 | 10.80 |
| 1.-3. Non-zero / low overlap (accepted) | 31,094,385 | 23.60 | 94.80 | 4.40 | 5.20 |
| 1.-3. Non-zero / low overlap (discarded) | 34,285,591 | 1.60 | 46.80 | 9.20 | 53.20 |
| 4.-5. Symbol+Language filter (accepted) | 26,609,214 | 25.00 | 97.20 | 2.80 | 2.80 |
| 4.-5. Symbol+Language filter (discarded) | 4,485,171 | 11.20 | 85.60 | 9.20 | 14.40 |
| 6. Similar pairs (accepted) | 4,666,464 | 12.00 | 86.80 | 12.80 | 13.20 |
| 7. Similar segments (accepted) | 2,081,354 | 14.80 | 95.60 | 3.60 | 4.40 |
| ParIce shallow filtering | | | | | |
| 0. ParIce 21.10 filtered | 1,776,049 | 73.60 | 95.20 | 4.80 | 4.80 |

Table 1: Size and manual evaluation results for the shallow filtering approaches. For each dataset 250 randomly sampled pairs are evaluated. 3C stands for all correct codes: CC, CB and CS. 3X stands for all error codes: X, WL and NL. For comparison, we also evaluate the clean version of the corpus as published by the ParaCrawl project. Note that we evaluated both accepted and discarded pairs for two of the filtering steps.

vector machine (Cortes and Vapnik, 1995) and logistic regression (Cox, 1958) classifiers.

**Sentence Perplexity using GPT-2**

Manual evaluation of ParaCrawl sentence pairs revealed that the Icelandic sentences in ParaCrawl are frequently ungrammatical or have erratic syntax, even though some, and in some cases most or all, of the lexical semantics of the translations are correct. This is likely because many web pages, scraped by the ParaCrawl project, use MT models to generate texts in multiple languages, even though the MT models do not generate fluent results. We try to find these badly formed sentences by training a classifier to recognize fluent and disfluent sentences. The classifier uses a pre-trained GPT-2 model (Radford et al., 2019), trained on the Icelandic Gigaword Corpus (Steingrímsson et al., 2018).[1] To train the classifier, we selected 10,000 sentences randomly from WikiMatrix (Schwenk et al., 2021) and ParaCrawl v8, and manually classified them in two groups: *coherent* (6,570 sentences) and *incoherent* (3,430 sentences).[2] The classifier uses the GPT-2 model to calculate perplexity for the sentences, and chooses potential thresholds as the average between two adjacent perplexity values. It then uses a maximization function to decide on a threshold that yields the most accurate prediction based on the training set.

### 3.3 Manual Evaluation

We manually annotated samples of the data sets compiled by each filtering approach. In our evaluation, we followed the taxonomy developed by Kreutzer et al. (2022), but slightly amended one category, CB, to include partial alignments. The taxonomy uses three codes for correct pairs and three error codes:

- CC – Correct translation, natural sentence.

- CB – Correct translation, boilerplate, partial alignments or grammatical errors.

- CS – Correct translation, short.

- X – Incorrect translation.

- WL – Either sentence in wrong language.

- NL – Either sentence is non-linguistic content.

**Shallow Filters:** We annotated 250 randomly selected pairs from the datasets at different stages of shallow filtering. Table 1 shows the size of the datasets after applying shallow filters, and the percentage of sentence pairs in different categories. The evaluation indicates that almost 70% of the raw ParaCrawl data is potentially useful, while over 30% is in the best case useless and possibly detrimental. Note that this describes the dataset before any filtering or deduplication has been carried out. ParaCrawl also distributes a cleaned version of the corpus, containing approximately $3M$ sentence pairs. In that version, over 10% of sentence pairs are still erroneous and, while almost

---

[1] The model, trained by Jón Friðrik Daðason, is available on Hugging Face: `https://huggingface.co/jonfd/gpt2-igc-is/tree/v1.0`.

[2] Dataset available here: `https://github.com/steinst/filter-align-datasets`

| | Laser | | | | | | | | LaBSE | | | | | | | |
|---|---|---|---|---|---|---|---|---|---|---|---|---|---|---|---|---|
| | ParaCrawl | | | | ParIce | | | | ParaCrawl | | | | ParIce | | | |
| | CC | 3C | X | 3X | CC | 3C | X | 3X | CC | 3C | X | 3X | CC | 3C | X | 3X |
| 0.0 ⇒ 0.1 | 10 | 100 | 0 | 0 | 95 | 100 | 0 | 0 | 0 | 7 | 93 | 93 | 1 | 9 | 91 | 91 |
| 0.1 ⇒ 0.2 | 9 | 99 | 1 | 1 | 93 | 99 | 1 | 1 | 0 | 5 | 95 | 95 | 4 | 12 | 88 | 88 |
| 0.2 ⇒ 0.3 | 8 | 99 | 1 | 1 | 92 | 100 | 0 | 0 | 1 | 6 | 94 | 94 | 11 | 26 | 74 | 74 |
| 0.3 ⇒ 0.4 | 16 | 100 | 0 | 0 | 87 | 100 | 0 | 0 | 0 | 7 | 93 | 93 | 14 | 50 | 50 | 50 |
| 0.4 ⇒ 0.5 | 16 | 99 | 1 | 1 | 83 | 99 | 1 | 1 | 2 | 13 | 85 | 87 | 24 | 75 | 25 | 25 |
| 0.5 ⇒ 0.6 | 20 | 85 | 14 | 15 | 75 | 98 | 2 | 2 | 4 | 42 | 57 | 58 | 46 | 93 | 7 | 7 |
| 0.6 ⇒ 0.7 | 15 | 69 | 31 | 31 | 61 | 90 | 10 | 10 | 16 | 71 | 29 | 29 | 64 | 98 | 2 | 2 |
| 0.7 ⇒ 0.8 | 10 | 43 | 57 | 57 | 58 | 93 | 7 | 7 | 26 | 94 | 6 | 6 | 82 | 100 | 0 | 0 |
| 0.8 ⇒ 0.9 | 13 | 56 | 42 | 44 | 63 | 75 | 25 | 25 | 15 | 98 | 2 | 2 | 89 | 99 | 1 | 1 |
| 0.9 ⇒ 1.0 | 27 | 63 | 36 | 37 | 51 | 65 | 36 | 36 | 11 | 99 | 1 | 1 | 99 | 100 | 0 | 0 |
| | NMTScore | | | | | | | | WAScore | | | | | | | |
| | ParaCrawl | | | | ParIce | | | | ParaCrawl | | | | ParIce | | | |
| | CC | 3C | X | 3X | CC | 3C | X | 3X | CC | 3C | X | 3X | CC | 3C | X | 3X |
| 0.0 ⇒ 0.1 | 22 | 76 | 24 | 24 | 65 | 92 | 8 | 8 | 1 | 17 | 81 | 83 | 8 | 45 | 55 | 55 |
| 0.1 ⇒ 0.2 | 20 | 96 | 4 | 4 | 87 | 100 | 0 | 0 | 12 | 46 | 53 | 54 | 43 | 91 | 9 | 9 |
| 0.2 ⇒ 0.3 | 12 | 98 | 2 | 2 | 85 | 100 | 0 | 0 | 28 | 72 | 21 | 28 | 57 | 95 | 5 | 5 |
| 0.3 ⇒ 0.4 | 9 | 100 | 0 | 0 | 94 | 100 | 0 | 0 | 27 | 88 | 9 | 12 | 73 | 97 | 3 | 3 |
| 0.4 ⇒ 0.5 | 9 | 100 | 0 | 0 | 97 | 100 | 0 | 0 | 39 | 96 | 4 | 4 | 80 | 100 | 0 | 0 |
| 0.5 ⇒ 0.6 | 12 | 99 | 1 | 1 | 97 | 100 | 0 | 0 | 33 | 95 | 5 | 5 | 92 | 100 | 0 | 0 |
| 0.6 ⇒ 0.7 | 13 | 100 | 0 | 0 | 93 | 100 | 0 | 0 | 27 | 93 | 7 | 7 | 93 | 99 | 1 | 1 |
| 0.7 ⇒ 0.8 | 11 | 99 | 0 | 1 | 99 | 100 | 0 | 0 | 10 | 99 | 1 | 1 | 94 | 99 | 1 | 1 |
| 0.8 ⇒ 0.9 | 23 | 100 | 0 | 0 | 100 | 100 | 0 | 0 | 7 | 97 | 3 | 3 | 94 | 99 | 1 | 1 |
| 0.9 ⇒ 1.0 | 20 | 100 | 0 | 0 | 100 | 100 | 0 | 0 | 5 | 98 | 2 | 2 | 95 | 100 | 0 | 0 |

Table 2: Results of the manual evaluation of samples of 100 randomly selected sentence pairs from each of ten bands for the scoring mechanisms used.

90% are potentially useful, only 13.6% are evaluated to be good mutual translations. We filter the raw data and evaluate the changes after in between shallow filtering steps. All the filters discard some mutual translations but proportionally more inadequate pairs. While the 3C column indicates the ratio of all pairs in the correct category, it includes boilerplate and ungrammatical segments not necessarily useful for MT. We want our filters to keep as many sentence pairs from the CC category and remove all from the X-categories. After filters 1-7 (see Section 3.2) have been applied, we see the number of pairs annotated as correct, CC, is 14.8%. After filter 5 this was 25%, but the last two filters lower the ratio because sentences that are identical, except for numbers or other named entities, have been reduced to one example.

We only manually evaluate the ParIce corpus after applying all the shallow filters and do not investigate the changes at each stage. This is because the data in the corpus all comes from known document sources and should not contain as much noisy data as ParaCrawl. We find that about 5% of sentence pairs in the corpus are erroneous, a number largely in line with the original ParIce paper, where the evaluation indicated that 3.5% of the alignments were bad, but we also find that only about three out of every four sentence pairs are mutual translations, with about 20% being accepted as correct while being imperfect in some way, usually due to misalignments.

**Scores:** After evaluating the shallow filters, we evaluated the effectiveness of the scoring mecha-

nisms. We divided the scores for each scoring approach into 10 bands, and manually evaluated 100 pairs for each band. The evaluation results, shown in Table 2, indicate that all the scoring methods have some merit and could probably be useful to a classifier. On their own, the results usually differ depending on the parallel corpora used, with the accuracy of the same scoring mechanism varying for different corpora. For example, the LaBSE score has to be more than 0.7 for more than 90% of sentence pairs in a scoring band to be acceptable (3C) for ParaCrawl, but only 0.5 for ParIce.

The distribution of the scores differ between scoring approaches, which can effect their usefulness. While NMTScore seems to be very accurate when looking at the bands in the table, 83% of the ParaCrawl sentences have a score of less than 0.3, and 25% of the ParIce sentences have a score of less than 0.1, indicating that even though the results seem very good, using only this scoring method may not be enough for accurate filtering. It should also be noted that most approaches do not seem to be very good at discerning finer nuances such as whether a sentence pair contains only mutual translations or if there is additional content in at least one of the sentences. The ratio of CC usually does not change as consistently with rising scores as the 3C or 3X ratio. This may indicate that if some sentence pairs classified as CB are detrimental to MT training, we need other approaches to identify them and filter out.

**Filters:** We annotated 100 pairs from each group of stochastic filtering approaches. We use the clas-

| Filter | ParaCrawl Filters | | | | | | | | | |
| | accepted (%) | | | | | rejected (%) | | | | |
| | No. pairs | CC | 3C | X | 3X | No. pairs | CC | 3C | X | 3X |
|---|---|---|---|---|---|---|---|---|---|---|
| GPT-2 | 1,218,256 | 15 | 93 | 7 | 7 | 863.098 | 5 | 91 | 8 | 9 |
| Logistic Regression | 1,940,385 | 38 | 85 | 4 | 15 | 140,969 | 18 | 37 | 61 | 63 |
| Random Forest | 1,981,405 | 7 | 98 | 0 | 2 | 99,949 | 2 | 22 | 77 | 78 |
| Support Vector Machine | 1,991,924 | 12 | 98 | 2 | 2 | 89,430 | 0 | 22 | 78 | 78 |
| Bicleaner baseline (0.50) | 1,973,885 | 22 | 96 | 4 | 4 | 107,469 | 10 | 41 | 58 | 59 |
| Bicleaner baseline (0.67) | 1,705,042 | 15 | 98 | 2 | 2 | 376,312 | 20 | 80 | 20 | 20 |
| Bicleaner retrained (0.50) | 1,898,209 | 25 | 97 | 3 | 3 | 183,145 | 27 | 75 | 25 | 25 |
| Bicleaner retrained (0.67) | 1,615,913 | 20 | 98 | 2 | 2 | 465,441 | 24 | 81 | 18 | 19 |
| Bicleaner lemmatized (0.50) | 1,850,884 | 18 | 88 | 8 | 12 | 230,470 | 14 | 66 | 28 | 34 |
| Bicleaner lemmatized (0.67) | 1,512,437 | 30 | 93 | 5 | 7 | 568,917 | 21 | 70 | 29 | 30 |
| Bicleaner AI (0.50) | 1,235,771 | **33** | **99** | 1 | 1 | 845,583 | 6 | 85 | 13 | 15 |
| Bicleaner AI (0.67) | 1,096,288 | 25 | 97 | 3 | 3 | 985,066 | 8 | 92 | 8 | 8 |
| Filter | ParIce filters | | | | | | | | | |
| | accepted (%) | | | | | rejected (%) | | | | |
| | No. pairs | CC | 3C | X | 3X | No. pairs | CC | 3C | X | 3X |
| GPT-2 | 1,444,956 | 81 | 96 | 4 | 4 | 331,093 | 68 | 91 | 9 | 9 |
| Logistic Regression | 1,560,346 | 85 | 100 | 0 | 0 | 215,703 | 49 | 77 | 23 | 23 |
| Random Forest | 1,667,847 | 86 | 99 | 1 | 1 | 108,202 | 20 | 51 | 49 | 49 |
| Support Vector Machine | 1,646,183 | 91 | 100 | 0 | 0 | 129,866 | 28 | 58 | 42 | 42 |
| Bicleaner baseline (0.50) | 1,546,216 | 85 | 99 | 1 | 1 | 229,833 | 35 | 79 | 21 | 21 |
| Bicleaner baseline (0.67) | 1,242,258 | 86 | 100 | 0 | 0 | 533,791 | 48 | 86 | 14 | 14 |
| Bicleaner retrained (0.50) | 1,499,610 | 85 | 99 | 1 | 1 | 276,439 | 42 | 90 | 10 | 10 |
| Bicleaner retrained (0.67) | 1,244,412 | 94 | 100 | 0 | 0 | 531,637 | 55 | 95 | 5 | 5 |
| Bicleaner lemmatized (0.50) | 1,463,780 | 89 | 100 | 0 | 0 | 312,267 | 50 | 90 | 10 | 10 |
| Bicleaner lemmatized (0.67) | 1,117,814 | 88 | 100 | 0 | 0 | 604,235 | 69 | 98 | 2 | 2 |
| Bicleaner AI (0.50) | 1,262,313 | **95** | **100** | 0 | 0 | 513,736 | 60 | 86 | 13 | 14 |
| Bicleaner AI (0.67) | 1,152,319 | 91 | 100 | 0 | 0 | 623,730 | 77 | 95 | 5 | 5 |

Table 3: Manual evaluation of datasets generated by different filtering approaches. We both evaluate sentence pairs accepted by each filtering approach, and rejected by it.

sifiers and Bicleaner models described in section 3.2. We set cutoff score at two different levels for each Bicleaner model, 0.5 and a higher threshold of 0.67 to try to discover whether detrimental sentence pairs can still be found at such a high level.

As evident in Table 3, the filtering mechanisms are quite adept at removing erroneous sentence pairs. We can see that for both corpora, all but two filters return over 90% accepted sentence pairs, and a low rate of erroneous data, and for ParIce, in particular, almost all erroneous data is removed for 8 out of 12 filtering approaches. However, as it is important to keep as many of the good sentence pairs as possible, the most useful approaches may be the ones that remove the fewest mutual translations. We see that while the Bicleaner AI model has the highest proportion of CC, mutual translations, it has the drawback of filtering out the highest proportion of sentences compared to almost all other approaches. Almost half of the ParaCrawl data was rejected, 985,066 out of 2,081,354 sentence pairs, when the threshold score is set to 0.67, of which 92% were rated in one of the correct categories. In order to investigate further what is best for MT training, we next train multiple models, using all the different data sets we have compiled, in order to see how the translations generated by these models compare to the results of our manual evaluation.

## 3.4 Automatic evaluation

We evaluate the effect of different filtering approaches on a downstream NMT task by training different MT models for each of the compiled datasets and evaluate them using BLEU. We use fairseq (Ott et al., 2019) to train Transformer$_{\text{BASE}}$ models, as described in Vaswani et al. (2017), except that we set dropout to 0.2 and use BPE with a shared vocabulary size of 32k. We train each model on a single A100 GPU and use early stopping with the patience set to 10 epochs on validation loss. We use the development and test sets provided for the English–Icelandic news translation task at WMT 2021 (Akhbardeh et al., 2021), using SacreBLEU.[3] Following Koszowski et al. (2021), we apply regular expressions to fix quotation marks post-translation, making sure Icelandic quotation marks are used in the Icelandic translations and English quotation marks in the English translations.

We train baseline models using the cleaned ParaCrawl dataset and the most recent published version of ParIce, and compare them to models trained on filtered datasets. Table 4 shows resulting BLEU scores. We used the pairwise bootstrap test (Koehn, 2004) to calculate statistical significance. Scores in bold are the highest, but not significantly higher than scores in italics. When models are trained with a cleaner dataset, they seem

---
[3]SacreBLEU Signature: BLEU+numrefs.1+case.mixed +tok.13a+smooth.exp+version.2.2.0

| Dataset | ParaCrawl training experiments | | | | | ParIce training experiments | | | | |
|---|---|---|---|---|---|---|---|---|---|---|
| | no. pairs | en→is | time | is→en | time | no. pairs | en→is | time | is→en | time |
| Baseline: ParaCrawl v9 clean | 2,967,519 | 20.2 | 23h3m | 30.6 | 11h14m | | | | | |
| Baseline: ParIce 21.10 | | | | | | 1,864,679 | 19.1 | 22h54m | 25.7 | 15h06m |
| Shallow filter 5 - Similar pairs | 4,666,464 | 19.1 | 18h9m | 30.4 | 29h56m | | | | | |
| Shallow filter 6 - Similar segments | 2,081,354 | 20.0 | 13h3m | 31.9 | 15h57m | | | | | |
| ParIce shallow filters | | | | | | 1,776,049 | *19.7* | 23h29m | 25.5 | 14h31m |
| IS-perplexity (GPT-2) | 1,218,256 | **21.1** | 5h50m | **33.0** | 14h11m | 1,444,956 | 18.5 | 22h33m | 24.7 | 10h18m |
| SVM | 1,991,924 | 19.6 | 13h41 | 32.4 | 15h56m | 1,646,183 | *19.8* | 17h38m | *26.0* | 13h04m |
| Logistic Regression | 1,940,385 | 20.1 | 11h48 | 32.1 | 12h01m | 1,560,346 | 19.2 | 16h51m | *26.1* | 13h30m |
| Random Forest | 1,981,405 | 19.5 | 6h37m | 31.8 | 15h32m | 1,667,847 | 18.6 | 20h07m | 25.2 | 12h22m |
| Bicleaner 1.5 (0.50) | 1,973,885 | 19.5 | 11h25m | 32.2 | 15h33m | 1,546,216 | *19.5* | 21h52m | **26.2** | 12h5m |
| Bicleaner 1.5 (0.67) | 1,705,042 | 19.3 | 8h29m | 31.4 | 8h53m | 1,242,258 | *19.5* | 12h06m | 25.6 | 9h01m |
| Bicleaner retrained (0.50) | 1,898,209 | 18.9 | 8h17m | 31.9 | 15h41m | 1,499,610 | *19.7* | 7h13m | 25.6 | 12h22m |
| Bicleaner retrained (0.67) | 1,615,913 | 19.5 | 7h36m | 30.5 | 12h59m | 1,244,412 | **19.8** | 10h16m | 25.5 | 6h13m |
| Bicleaner lemmatized (0.50) | 1,850,884 | 19.6 | 10h29m | 31.6 | 17h19m | 1,463,780 | **19.8** | 15h12m | 25.9 | 11h56m |
| Bicleaner lemmatized (0.67) | 1,512,437 | 19.3 | 6h27m | 30.9 | 8h32m | 1,171,814 | **19.8** | 7h29m | 25.6 | 8h56m |
| Bicleaner AI (0.50) | 1,235,771 | 20.5 | 8h26m | 31.7 | 7h15m | 1,262,313 | 19.1 | 7h07m | *26.1* | 7h44m |
| Bicleaner AI (0.67) | 1,096,288 | *21.0* | 4h50m | 30.8 | 3h45m | 1,152,319 | 18.9 | 7h11m | 25.1 | 7h28m |

Table 4: BLEU scores and training time for different filtering approaches. Scores in bold are the highest for the dataset and translation direction. Scores in italics are lower, but not significantly lower than the highest ones ($p > 0.05$).

to converge faster, even though the model quality is the same or better. We know from our manual evaluation that most of these training sets contain some erroneous pairs, and in order to try to reduce the number of these, we select the dataset resulting in the highest BLEU score out of the datasets compiled by a Bicleaner model and the best resulting dataset compiled by a classifier. We do an ablation study to investigate whether combining these filters, and the filter looking at perplexity in Icelandic sentences, leads to a better training set. For each dataset and translation direction, we combine the highest scoring Bicleaner model with combinations of the highest scoring statistical classifier and the GPT-2 classifier. Table 5 shows the results for different combinations. For the English→Icelandic translation direction, we obtain higher scores for both corpora using a combination, but for Icelandic→English the BLEU scores never exceed the best standalone filters. We speculate this may be due to noise being more common in the Icelandic texts instigating a need for more filtering when Icelandic is the target language, making the combined filters a better choice in that case, but further investigation is needed.

For our final models, we concatenate the highest-scoring datasets from ParaCrawl and ParIce. These models obtain the highest BLEU

| ParaCrawl en→is filters | | | |
|---|---|---|---|
| Dataset | no. pairs | BLEU | time |
| Bicleaner AI (0.67) + LogReg | 1,071,802 | 20.4 | 3h51m |
| Bicleaner AI (0.67) + GPT-2 | 776,984 | **21.5** | 4h17m |
| Bicleaner AI (0.67) + LogReg + GPT-2 | 756,503 | 20.7 | 3h40m |
| ParaCrawl is→en filters | | | |
| Dataset | no. pairs | BLEU | time |
| Bicleaner 1.5 (0.50) + SVM | 1,930,998 | 32.3 | 20h24m |
| Bicleaner 1.5 (0.50) + GPT-2 | 1,147,961 | 31.9 | 9h02m |
| Bicleaner 1.5 (0.50) + SVM + GPT-2 | 1,119,400 | 32.1 | 7h32m |
| ParIce en→is filters | | | |
| Dataset | no. pairs | BLEU | time |
| Bicleaner Lemmatized (0.50) + SVM | 1,405,446 | **20.2** | 17h59m |
| Bicleaner Lemmatized (0.50) + GPT-2 | 1,205,070 | 19.6 | 14h04m |
| Bicleaner Lemmatized (0.50) + SVM + GPT-2 | 1,161,337 | 18.9 | 13h24m |
| ParIce is→en filters | | | |
| Dataset | no. pairs | BLEU | time |
| Bicleaner 1.5 (0.50) + LogReg | 1,430,015 | *26.1* | 13h22m |
| Bicleaner 1.5 (0.50) + GPT-2 | 1,269,808 | 25.7 | 9h30m |
| Bicleaner 1.5 (0.50) + LogReg + GPT-2 | 1,179,158 | 25.7 | 10h46m |
| Best datasets from both corpora combined | | | |
| Dataset | no. pairs | BLEU | time |
| is→en: ParaCrawl – GPT-2 + ParIce Bicleaner 1.5 (0.50) | 2,764,472 | ***33.2*** | 15h55m |
| en→is: ParaCrawl – Bicleaner AI (0.67) + GPT-2 + ParIce – Bicleaner Lemmatized (0.50) + SVM | 2,182,430 | ***22.6*** | 18h57m |

Table 5: BLEU scores and training time for combinations of different filtering approaches. While datasets compiled with combined filters were used to train MT systems delivering the highest BLEU scores for the English→Icelandic translation direction, for Icelandic→English the highest scoring systems were trained on data compiled with only one filter. Scores in bold are the highest scores for the dataset and translation direction they represent. Scores in italics are lower, but not significantly lower ($p > 0.05$). Scores in bold and italics are the highest scores obtained for the translation direction.

scores, 33.2 for Icelandic→English and 22.6 for English→Icelandic. We compare these scores to the results of systems submitted to the WMT 2021 news translation task for the same language pair and directions. Koszowski et al. (2021) submitted a system trained on ParIce and ParaCrawl as well as WikiMatrix and wikititles. The model, based on Transformer$_{BIG}$ (Vaswani et al., 2017), using back-translation and forward-translation for data augmentation, achieved 22.7 and 33.3 BLEU for en→is and is→en, respectively, only slightly higher, and probably not significantly higher, than our best scores. Símonarson et al. (2021) trained MT models using mBART-25, employing 16 V100 GPUs. They employed back-translations in their training and achieved 22.7 and 32.9 BLEU for en→is and is→en, respectively, after training for 4 days, and after another 4 days and adding more back-translations, they reached 24.3 and 33.5 for en→is and is→en, respectively. These are slightly higher than our best scores. However, we only filter, while they use data augmentation, larger models and more computing power for much longer periods of time.

## 4 Conclusions and Future Work

In regards to our research questions, our results indicate that different filtering approaches suit different datasets and translation directions, even though we are working within the same language pair. Manual inspection of filtering results and scoring mechanisms seem to be helpful for making informed decisions on how best to filter a dataset. For best results, filtering approaches should be chosen for each translation direction. A limitation of our work is that it does not show which data are detrimental and which are beneficial. In future work, we want to investigate if the differences between datasets used for training can give us an idea of which sentence pairs are most important to filter out. We intend to do this by investigating the pairs discarded by our filters, to compare the data that leads to rising BLEU scores and that which lowers them. This could lead to insights that help constructing filters that work on a more fine-grained level when that is needed.

Our manual evaluation shows that the scores, generated by the automatic scoring systems we employ, have different interpretations depending on the dataset. If scores are used for filtering parallel data, the optimal score should lead to a

dataset that produces the best MT model. Feng et al. (2022) suggest a threshold of 0.6 for LaBSE when mining parallel text from CommonCrawl, stating that the threshold was selected by manually inspecting sampled data, but do not specify the language pair used when inspecting the data. In order for the scoring mechanism to be most effective, the user should inspect the results for their dataset before setting a threshold. While all our scoring mechanisms seem to be useful, none of the methods are very good at identifying mutual translations in particular, labelled CC in our taxonomy.

We trained two Bicleaner models for our experiments and our lemmatized model gave the best results for filtering ParIce for the en→is translation direction. The Bicleaner models could perhaps be improved. Bicleaner uses n-gram models and we only used a part of our parallel corpora to train these. If we would use larger corpora the n-gram models would likely give us more accurate scores. The bilingual probability dictionary we used only contained lemmas. By producing all wordforms for the lemmas and trying to estimate the prevalence of each wordform using a monolingual corpus, we could perhaps provide an unlemmatized model with more accurate information leading to better results. Furthermore, we only use 10k sentences to train our GPT-2 perplexity model for Icelandic. A larger dataset may increase its accuracy.

Two systems participating in the WMT 2021 news translation task, evaluated on the same data, obtain scores only slightly higher than ours, but while we only train a Transformer$_{BASE}$ model, they train larger models using more resources and much longer training time. In our experiments, models that have been filtered more tend to converge faster. We can deduce from this that training data that is better filtered, not only improves MT output quality, but is also in line with a call to greener and more sustainable models of AI, see e.g. Yusuf et al. (2021) and Jooste et al. (2022).

## Acknowledgements

This work was supported by the The Icelandic Centre for Research, RANNIS grant number 228654-051, and by the ADAPT Centre for Digital Content Technology which is funded under the Science Foundation Ireland (SFI) Research Centres Programme (Grant No. 13/RC/2106) and is co-funded under the European Regional Development Fund.

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
