# OpenReview forum: "Filtering Matters: Experiments in Filtering Training Sets for Machine Translation"
_NoDaLiDa/2023/Conference — NoDaLiDa 2023_

### Official Review · Reviewer_hFPV · 2023-02-19
**Work on parallel data filtering for English-Icelandic with both manual and automatic evaluation of both parallel data filtering and the downstream neural machine translation tasks.**

**Rating:** 8
**Confidence:** 5

**Review:**


The authors present their work on parallel data filtering for English-Icelandic. The authors analyse what filtering methods suit better for each translation direction. They also train several classifiers and perform manual and automatic evaluation both for the filtering task and then the downstream machine translation task. The authors show that filtering is a challenging problem that has not yet been solved, and it is important to carry out research on it. The authors compare various filtering methods and show that there is no one good method that can identify smaller divergences from parallelity. It is yet to be identified (in future work) how much those smaller divergences affect translation quality. All in all, I believe that the authors show a strong contribution (although it is difficult not to get lost in the numerous experiments and the results), and the paper could be a good fit for the conference.

Questions:

1) ParaCrawl (at least for English-Latvian) contains quite a lot of machine-translated content. Some of it is correct but ungrammatical. Where in your evaluation classification would this fall into? CB? Do you consider the machine-translated pairs as good or bad?

2) When working on WMT shared tasks (some years ago), we noticed that the machine-translated content (i.e., ParaCrawl) improves quality for systems that translate into English but lowers quality for systems that translate out of English. It was expectable since the machine-translated content worked as back-translated data where the source was not necessarily grammatical. Do you see a similar tendency for Icelandic->English vs. English->Icelandic? Just a hypothesis that I have is that the different filtering methods are needed to handle ungrammaticality and noise on the target side. Fruit for thought!

3) It is unclear how the authors decided to set the Bicleaner threshold to 0.5 and 0.67. These seem to be some arbitrary random values! More reasoning would be needed.

4) "early stopping with the patience set to 10 epochs" - is "epochs" a typo here? Typically early stopping is performed after X negative validations, where each validation is performed after Y updates (e.g., 1000 updates). Dataset sizes may warry a lot. What you work on is rather small, and, in your case, early stopping on epochs may be ok, but it will definitely not be for 20M, 50M, or 100M (etc.) datasets. For those, having a 10-epoch early stopping criterion may require training the system for an unnecessarily long time!

5) It is difficult to not drown in the results. Consider somehow highlighting the best results. I.e., What would you recommend when training systems into English and out of English?

Language usage mistakes:

022 - plural needed.

036 - subject-verb disagreement.

075 - reference missing to Bicleaner (it may or may not be defined somewhere further, but this is the first time Bicleaner is mentioned!)

074-075 - "We compare these [filtering methods] ... to the two corpora". The sentence does not make sense! What does it mean to compare a filtering method to a corpus? Those are two non-comparable things.

500 - "reduced" - bad lexical choice (the wrong word in the given context).

760, 805 - missing commas after introductory phrases.

Formatting issues:

1) Table 1 is misplaced. It should be placed on page 5 since its mention is (only) on page 5.

2) Table 2 is misplaced. It should be placed on page 6.

3) 719 - space missing.


**Paper Type:**

Long paper

---

### Official Review · Reviewer_QeGN · 2023-03-08
**Nice paper on filtering parallel corpora for Machine Translation between English and Icelandic**

**Rating:** 9
**Confidence:** 4

**Review:**

This is a nice paper that reports on experiments with various filtering methods to optimize the parallel training corpus for NMT between English and Icelandic (in both directions). The experiments are thoroughly described.

Unfortunately the results are not breath-taking (as the authors themselves admit in the abstract). The most interesting finding is that filtering leads to shorter training times (based on reduced size of the training corpus) without (much) loss in BLEU scores. So, targeted filtering is a viable method towards "greener" NMT.

I like the paper and find hardly anything to complain about.

It would be nice to see some examples of EN - Icelandic sentence pairs that have been filtered out (by multiple methods) even though the authors deem these pairs to be well translated sentences.

Even more important: It would be nice to discuss what the results mean for other language pairs (in particular pairs that include Scandinavian languages).

There are a number of incomplete entries in the reference list. :-(
For example:
Alec Radford et al. --> year and publication venue missing;
Jannis Vamvas and Rico Sennrich. 2022. --> publication venue missing



**Paper Type:**

Long paper

---

### Official Review · Reviewer_hWnH · 2023-03-13
**good work**

**Rating:** 8
**Confidence:** 3

**Review:**

The paper presents a study on filtering strategies for building more reliable training data for machine translation. The authors focused on the Enlgish-Icelandic language pair, and on two common web-crawled parallel corpora, ParaCrawl and ParIce. The authors run a series of filtering strategies, from the lexical level, to using pre-trained neural representation, manually assessing a portion of the data for each configuration. Finally, they trained NMT models using the different filtered data, comparing also without using them. Together with bleu score, the authors show training time, useful to assess the amount of time required by the model to converge. Training on the best combination of filtered data results in performance that match a bigger system trained on more data.

Following a missing relevant citation, to include in the camera ready of the paper:
Aulamo, Mikko, Sami Virpioja, and Jörg Tiedemann. "OpusFilter: A Configurable Parallel Corpus Filtering Toolbox." Proceedings of the 58th Annual Meeting of the Association for Computational Linguistics: System Demonstrations. 2020.

**Paper Type:**

Long paper

---

### Decision · Program_Chairs · 2023-03-17

Accept